# Dopamine Transporter Knockout Rats Display Epigenetic Alterations in Response to Cocaine Exposure

**DOI:** 10.3390/biom13071107

**Published:** 2023-07-12

**Authors:** Samara Vilca, Claes Wahlestedt, Sari Izenwasser, Raul R. Gainetdinov, Marta Pardo

**Affiliations:** 1Department of Psychiatry and Behavioral Sciences, Miller School of Medicine, University of Miami, Miami, FL 33136, USA; svilca@miami.edu (S.V.); clawah@gmail.com (C.W.); sizenwasser@med.miami.edu (S.I.); 2Center for Therapeutic Innovation, Miller School of Medicine, University of Miami, Miami, FL 33136, USA; 3Institute of Translational Biomedicine, St. Petersburg University Hospital, St. Petersburg State University, Universitetskaya Emb. 7-9, 199034 St. Petersburg, Russia; gainetdinov.raul@gmail.com; 4Department of Neurology, Miller School of Medicine, University of Miami, Miami, FL 33136, USA

**Keywords:** cocaine, DAT, knockout, epigenetics, KDM6B, BRD4

## Abstract

(1) Background: There is an urgent need for effective treatments for cocaine use disorder (CUD), and new pharmacological approaches targeting epigenetic mechanisms appear to be promising options for the treatment of this disease. Dopamine Transporter (DAT) transgenic rats recently have been proposed as a new animal model for studying susceptibility to CUD. (2) Methods: DAT transgenic rats were treated chronically with cocaine (10 mg/kg) for 8 days, and the expression of epigenetic modulators, Lysine Demethylase 6B (KDM6B) and Bromodomain-containing protein 4 (BRD4), was examined in the prefrontal cortex (PFC). (3) Results: We show that only full knockout (KO) of DAT impacts basal levels of KDM6B in females. Additionally, cocaine altered the expression of both epigenetic markers in a sex- and genotype-dependent manner. In response to chronic cocaine, KDM6B expression was decreased in male rats with partial DAT mutation (HET), while no changes were observed in wild-type (WT) or KO rats. Indeed, while HET male rats have reduced KDM6B and BRD4 expression, HET female rats showed increased KDM6B and BRD4 expression levels, highlighting the impact of sex on epigenetic mechanisms in response to cocaine. Finally, both male and female KO rats showed increased expression of BRD4, but only KO females exhibited significantly increased KDM6B expression in response to cocaine. Additionally, the magnitude of these effects was bigger in females when compared to males for both epigenetic enzymes. (4) Conclusions: This preliminary study provides additional support that targeting KDM6B and/or BRD4 may potentially be therapeutic in treating addiction-related behaviors in a sex-dependent manner.

## 1. Introduction

Cocaine continues to be a powerful addictive substance with an urgent need for new therapeutical approaches for its treatment worldwide. Dopamine (DA) continues to be a focus of research, due to its role in controlling several cocaine-related behaviors, including locomotor sensitization, motivation, and reward [1]. More precisely, the DA transporter (DAT), which plays an important role in regulating intracellular and extracellular DA levels [2,3], has been studied widely for its role in various neurological diseases, including cocaine use disorder (CUD) [4,5,6,7]. Moreover, previous studies using DAT Knockout (KO) mice [8,9,10] and a recent study from our group using DAT KO rats [11] support the hypothesis that DA levels and the DAT mediate response to cocaine, as well as highlighting the impact of sex. Additionally, excessive DA signaling has been related to altered gene expression [12].

Numerous studies have linked drug-induced gene expression changes to epigenetic machinery [13]. Chromatin remodeling, specifically histone posttranslational modifications (PTMs), contribute to cocaine-related behaviors, as they affect the expression of specific genes in several brain areas related to reward [14,15,16,17,18]. Furthermore, epigenetic alterations may contribute to individual vulnerability to the rewarding effects of cocaine, which could lead to maladaptive and addictive behaviors [19,20]. Finally, epigenetic pharmacotherapies reduce cocaine-related behaviors and cocaine-induced transcriptional changes [21,22,23].

Recently, the histone lysine demethylase, KDM6B, has been implicated in cocaine-related behaviors. KDM6B is a JmjC domain-containing protein that directly opposes the Polycomb repressive complex 2 (PRC2) by demethylating Histone 3 Lysine 27 (H3K27me3) [24] and is involved in transcriptional elongation [25]. Furthermore, KDM6B has been shown to regulate several genes involved in development, neurogenesis, and neural plasticity due to its activity at the promoter and enhancer regions [26,27,28]. Interestingly, KDM6B expression is increased following cocaine-associated memory retrieval in the dorsal hippocampus [29], as well as during withdrawal and retrieval in the medial prefrontal cortex (PFC) of rats and mice [30,31]. Specifically, Li et al. [30] demonstrated changes in KDM6B expression related to cocaine chronic exposure, reporting an increase in KDM6B RNA levels 7 days after chronic cocaine exposure in mice. Additionally, KDM6B expression was elevated on the third day of cocaine extinction in male rats [31]. Altogether, these data implicate KDM6B as a relevant enzyme that should be considered when developing new treatments for CUD.

Other epigenetic enzymes, in particular the bromodomain and extraterminal domain (BET) family of proteins (BRD2, BRD3, BRD4, and BRDT), also have been shown to play an important role in mediating cocaine-related behaviors and molecular changes [21,32]. The BET family contains two bromodomains, which bind acetylated histone tails and are involved in transcriptional coactivation and elongation [33,34,35]. Specifically, BRD4 has been involved in a variety of cellular processes, such as cell proliferation, apoptosis, and transcription [36,37]. Indeed, BRD4-bound sites are highly correlated with increased gene expression [38] and are associated with super-enhancer function [28,39]. Several groups have begun to study BRD4 for its role in cocaine-related behavior [22,40], as histone acetylation has been shown to regulate cocaine-induced neuroadaptations [41]. More specifically, BRD4 protein and mRNA expression are increased in the nucleus accumbens (NAc) after cocaine exposure in male mice and rats [21,32]. Furthermore, our groups identified BRD4 as an essential component of the potentiated expression of brain-derived neurotrophic factor (BDNF) and memory [42], which play an important role in the development of substance use disorders (SUDs) [43].

Selective, small-molecule inhibitors of both KDM6B [30,44] and BRD4 [45,46,47] have been developed and have increased interest in their therapeutic utility [48,49,50,51,52,53,54]. However, the role of these epigenetic enzymes and their inhibition in CUD has only recently begun [22,32,40,42]. For example, the KDM6B inhibitor, GSK-J4, prevented the cocaine reconsolidation of cocaine memory and cocaine-primed reinstatement in a conditioned place preference (CPP) paradigm in male mice [23]. The selective BRD4 inhibitor, JQ1, also reversed the effects of cocaine on CPP in male mice [22,32,40,42].

Previous studies examining cocaine-related behaviors and their underlying epigenetic machinery have focused on the striatum and NAc due to their important role in addictive behavior and reward [55,56]. However, the prefrontal cortex (PFC) is also important to the development of SUD, as this region controls decision-making and seeking behavior [57]. Specifically, glutamate release via projections from the medial PFC to NAc contribute to aspects of drug-conditioned responses, such as drug-seeking and intake, during relapse as well as craving [58,59,60,61]. For instance, impulsivity and reward-seeking are personality traits that increase vulnerability to SUD and are thought to be controlled by the PFC [62,63]. Furthermore, cocaine impairs several PFC-dependent functions, including working memory [64]. In the current study, we investigated the expression of KDM6B and BRD4 in the PFC of male and female DAT transgenic rats in response to chronic cocaine administration using qRT-PCR. We hypothesize that KDM6B and BRD4 may play a vital role in mediating the heightened response to cocaine in DAT HET and KO rats as well as the impact of sex differences [11].

## 2. Materials and Methods

### 2.1. Animals

Wistar-Han Slc6a3 (DAT) KO rats were generated previously [65] and kindly provided by Dr. Gainetdinov. Rats were genotyped on postnatal day (PND) 18–21, as previously described [65,66,67]. Animals from both sexes and the three different genotypes (wild-type DAT +/+ (WT), heterozygous DAT +/− (HET) and homozygous DAT −/− (KO)) were used. The total number of animals used in the current study was N = 48. The number of animals in the Saline group was as follows: WT Male (*n* = 2), HET Male (*n* = 3), KO Male (*n* = 6); WT Female (*n* = 6), HET Female (*n* = 5), KO Female (*n* = 3). The number of animals in the Cocaine group was as follows: WT Male (*n* = 3), HET Male (*n* = 4), KO Male (*n* = 4); WT Female (*n* = 4), HET Female (*n* = 5), KO Female (*n* = 3). The average weights for the animals used prior to treatment were as follows: Male WT: 148 g; HET: 142 g; KO: 139 g; Female: WT: 127 g, HET: 132 g; KO: 122 g.

The study was conducted with rats aged PND42, in accordance with the guidelines of the Guide for the Care and Use of Laboratory Animals (Eighth Edition), National Research Council, Department of Health, Education and Welfare (2011). All studies were approved by the University of Miami Institutional Animal Care and Use Committee (Protocol 17–016).

### 2.2. Treatments

Cocaine hydrochloride was obtained from the National Institute on Drug Abuse (Rockville, MD, USA). Cocaine was administered intraperitoneally (ip) for 8 days. The cocaine dose (0 or 10 mg/kg) was chosen based on previous publications [11,68,69,70]. Saline 0.9% solution was used as vehicle.

### 2.3. Tissue Collection

At PND42, rats were euthanized by rapid decapitation 60 min after treatment (N = 2–6 per genotyping group). The PFC (Bregma 5.16–2.28 mm, Paxinos and Watson “The Rat Brain in Stereotaxic Coordinates”) was dissected (olfactory bulb was removed at approximately Bregma 5.16 mm) and flash-frozen using liquid nitrogen. The average tissue weight was 50–75 mg. Tissue was stored at −80 °C until processing.

### 2.4. qRT-PCR Analyses

Frozen rat tissue was thawed on ice and homogenized in 1 mL of TRIzol. An RNEasy Mini Kit (Qiagen, Germany) was used for RNA extraction according to the manufacturer’s instructions. RNA concentrations were determined using a NanoDrop 2000 (Thermo Fisher Scientific, Waltham, MA, USA). The average 260/280 and 260/230 ratios were 2.0 and 1.6, respectively. Total RNA was reverse-transcribed using a High-Capacity cDNA Reverse Transcription Kit (Thermo Fisher Scientific, Waltham, MA, USA). Using validated TaqMan primer probes for BRD4 (Thermo Fisher Scientific, Waltham, MA, USA, Rn01535560_m1) and KDM6B (Rn01471506_m1), cDNA was run in triplicate on the QuantStudio 6 Flex (Thermo Fisher Scientific, Waltham, MA, USA) and analyzed using the 2^−ΔΔCT^ method [71] with GAPDH (Rn01775763_g1) as a normalization control [72].

### 2.5. Statistical Analysis

All data were analyzed using GraphPad Prism (GraphPad Software version 9.0, Boston, MA, USA). Three-way ANOVAs (Treatment × Genotype × Sex) were performed for each enzyme. Following significant interactions, two-way ANOVAs (Treatment × Genotype) within each sex were performed to determine differences in gene expression. Two-way ANOVAs (Sex × Genotype) were performed to determine differences in gene expression at baseline. Following a significant interaction, post-hoc analysis was conducted using Bonferroni’s multiple comparisons test to further assess differences between groups. Statistical values of *p* ≤ 0.05 were considered significant.

## 3. Results

### 3.1. Analyses of the Epigenetic Marker KDM6B in DAT HET and KO Rats after Chronic Cocaine Exposure

KDM6B (Figure 1) and BRD4 (Figure 2) expressions were measured in the PFC of rats administered eight daily injections of cocaine (10 mg/kg/day) or saline, and tissues were removed one hour after the last injection. The preliminary qRT-PCR three-way ANOVA showed a significant effect of genotype (F(2,36) = 4.11, *p* = 0.0248), sex (F(1,36) = 47.60, *p* < 0.0001) and treatment (F(1,36) = 36.74, *p* < 0.0001). Examining these data further revealed that drug-naïve DAT KO female rats had a significantly elevated level of KDM6B expression compared to WT or HET female or male rats (two-way ANOVA F(2,18) = 6.48, *p* = 0.0076, Figure 1A). The levels of KDM6B did not differ in drug-naïve male rats between genotypes. However, drug-naïve female KO rats showed increased KDM6B basal levels compared to KO males as well as WT and HET females (Figure 1A).

In males, there was a significant interaction between genotype and treatment (two-way ANOVA F(2,16) = 11.70, *p* = 0.0007), revealing that KDM6B expression was significantly decreased in HET male rats after cocaine administration compared to saline (Figure 1B). Interestingly, post-hoc analysis revealed that within cocaine treatment, DAT KO rats had a significantly increased KDM6B expression compared to DAT HET rats (Figure 1B). In females, there was a significant effect of treatment (two-way ANOVA F(1,20) = 36.37, *p* < 0.0001), as repeated cocaine administration tended to increase expression across all genotypes. Specifically, in response to cocaine, an increasing trend could be observed in the WT group, and there was a significant increase in KDM6B expression in HET and KO rats (Figure 1C). Interestingly, there was trend where KO rats showed a greater increase in KDM6B expression compared to WT or HET (Figure 1C).

### 3.2. Analyses of the Epigenetic Marker BRD4 in DAT HET and KO Rats after Chronic Cocaine Exposure

The preliminary qRT-PCR three-way ANOVA for BRD4 also showed a significant effect of genotype (F(2,36) = 3.91, *p* = 0.0292), sex (F(1,36) = 28.20, *p* < 0.0001) and treatment (F(1,36) = 43.16, *p* < 0.0001). In contrast to KDM6B, BRD4 expression at baseline was not significantly different across sexes or genotypes (two-way ANOVA F(2,19) = 1.75, *p* = 0.1999, Figure 2A). However, the response to repeated cocaine administration was similar to the observed differences in KDM6B expression between males and females. Importantly, within sex, there was a significant interaction between genotype and treatment in males (two-way ANOVA F(2,16) = 55.98, *p* < 0.0001). Post-hoc analysis revealed that there was a significant increase in KDM6B expression in male KO but a significant decrease in WT and HET rats compared to drug-naïve animals (Figure 2B). Additionally, post-hoc analysis showed that within the cocaine treatment, DAT KO rats exhibited increased BRD4 expression compared to WT and HET rats (Figure 2B). In females, there was a significant effect of treatment (two-way ANOVA F(1,20) = 36.89, *p* < 0.0001), as repeated cocaine administration showed an increasing trend in the WT group and a significantly increased BRD4 expression in HET and KO rats (Figure 2C).

## 4. Discussion

Our current preliminary data highlight the relevance of epigenetic mechanisms in response to cocaine. We explored the impact of DAT genetic manipulations on the effect of cocaine exposure for two epigenetic enzymes that are implicated in cocaine-related behaviors. We used a recently developed DAT transgenic rat [65], an animal model of hyperdopaminergia that showed increased extracellular levels of DA in DAT HET and DAT KO rats in several brain areas (~2.5–3- and 7.5–8-fold, respectively, compared to control WT rats) [65,73]. Specifically, we studied the effects of chronic cocaine administration on mRNA expression levels of the histone lysine demethylase, KDM6B, as well as the BET protein, BRD4. We observed a strong effect of cocaine on these epigenetic enzymes in a sex- and genotype-dependent manner. These results support our previous study [11], which showed robust sex differences in response to cocaine administration in DAT transgenic rats. Additionally, these data highlight the impact of partial or full DAT mutation on epigenetic enzymes and subsequent effects on cocaine-related behavior.

KMD6B has been the focus of several recent studies due to its role in altering chromatin structures and regulating gene expression in response to cocaine [23,30,31]. In the current study, we evaluated KDM6B expression levels in the PFC of DAT transgenic rats. At baseline, DAT mutant rats showed a significantly elevated KDM6B expression in female KO rats compared to the other genotypes and to KO males. These data suggest that DAT mutations have a sex-dependent impact on KDM6B. Full mutation of DAT in female rats alone was strong enough to induce KDM6B expression, highlighting the importance of DAT in DA homeostasis and gene expression. It was shown that DA levels are increased in the PFC of DAT KO mice [72], and it is likely that it is mediated by an increased DA transmission in this brain area. Interestingly, previous studies have shown that KDM6B gene expression and protein levels are substantially increased after 7 days of withdrawal from cocaine in male mice [23,30], as well as on the third day of extinction after cocaine self-administration in male rats [31]. In our study, we collected tissue 60 min after the final cocaine injection, representative of early withdrawal, which could explain why we were unable to observe significant differences in KDM6B expression in the WT rats of both sexes. From our knowledge, we are the first to study the interaction of DAT mutations and sex on KDM6B expression levels in response to cocaine exposure. In female rats, cocaine administration increased KDM6B expression across all genotypes, with a higher trend in KO rats. On the other hand, male rats showed a tendency to reduce KDM6B expression, especially in rats with a partial DAT mutation (HET). Further work is required to understand the underlying mechanisms governing these effects.

BRD4, an epigenetic reader protein, participates in cocaine-induced reward and neuroplasticity [32]. In drug-naïve rats, genetic DAT alterations do not impact BRD4 expression levels in both sexes. As observed with KDM6B expression, cocaine produces opposing effects across sexes. Previous studies report that cocaine increases the expression of BRD4 in the NAc and striatum of WT animals [21,32,42]. Our data expand on these previous data by determining the impact of DAT and DA transmission on chronic cocaine exposure in both males and females. Specifically, chronic cocaine administration reduced BRD4 expression in the PFC of male WT and HET rats. However, cocaine induced a robust increase in BRD4 expression in KO males. Furthermore, as observed with KDM6B expression, female HET and KO rats showed a significant increase in BRD4 expression levels after chronic exposure to cocaine, an effect that only showed a trend in WT female rats. Previous studies have determined the effect of cocaine on the NAc and striatum of male animals, as these brain regions are integral to regulating cocaine’s rewarding effects [21,32,42]. Since cocaine cannot affect DAT-mediated processes in DAT KO mice [5,10,65], it is likely mediated by other neurotransmitter systems affected by cocaine, such as serotonin. The significant sex differences demonstrated by our data warrant further study. Additionally, different brain regions that contribute to the development of CUD must be considered when interpreting current and past data. Of note, it is important to discuss the magnitude of the effects that we see in females compared to male rats. The previous literature indicates that there are numerous sex differences in response to cocaine exposure [74,75,76,77,78]. However, the mechanisms underlying sex-dependent effects have not been fully elucidated. Here, we show that KDM6B and BRD4 may be novel targets for treating CUD. The expressions of both epigenetic enzymes in response to cocaine administration were 20–40-fold (KDM6B) and 4–6-fold (BRD4) higher in female DAT transgenic rats, indicating that cocaine had a greater impact on females. This lends support to the sex-dependent behavioral changes observed in a recent publication by our group using the same transgenic rats [11].

We acknowledge the limitations of our study. Future studies will include several treatment conditions (acute versus chronic; several cocaine doses; several collection timepoints) and will compare exposure to cocaine during adolescence and adulthood. Further sex differences will be considered, including the study of hormones’ implication in current variables.

Altogether, our data suggest that DAT plays an important role in regulating the behavioral and biochemical response to cocaine, including but not limited to epigenetic changes. Additionally, the robust sex differences observed underscore the need for further studies examining both sexes. Previous studies have shown that inhibition of KDM6B and BRD4 reduced the rewarding effects of cocaine [21,22,23,32,42]. Our current study is in concert with previous publications that show the likely importance of these epigenetic changes [13,21,22,52] and strongly supports the pursuit of epigenetic enzymes as new therapeutic targets for SUD.

## 5. Conclusions

We conclude, in agreement with our recent publication [11], that the response to cocaine depends on DAT expression levels and that epigenetic mechanisms may play a fundamental role in sex differences. Current preliminary data provide information on the relationship between DAT levels and the epigenetic enzymes KDM6B and BRD4 as well as on their interaction after cocaine exposure. Altogether, this study indicates that KDM6B and BRD4 activity may be a possible mechanism underlying CUD-related behaviors (such as drug-seeking, craving, reward, etc.). Future studies using selective inhibitors for both enzymes will be necessary to determine if these epigenetic enzymes could be proposed as future novel therapeutic targets for SUD.

## Figures and Tables

**Figure 1 biomolecules-13-01107-f001:**
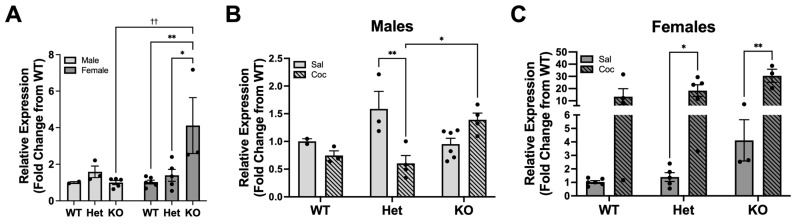
KDM6B gene expression levels in the PFC of DAT transgenic rats. (**A**) Basal level of KDM6B across sexes and genotypes ((Saline Males: WT (*n* = 2), HET (*n* = 3), KO (*n* = 6); Saline Females: WT (*n* = 6), HET (*n* = 5), KO (*n* = 3); Cocaine Males: WT (*n* = 3), HET (*n* = 4), KO (*n* = 4); Cocaine Females: WT (*n* = 4), HET (*n* = 5), KO (*n* = 3)). * *p* < 0.05, ** *p* < 0.01, significant difference between genotypes; ^††^
*p* < 0.01, significant difference between sexes. Effect of cocaine on KDM6B gene expression across (**B**) male and (**C**) female rats. * *p* < 0.05, ** *p* < 0.01 represent significant differences. Note different y-axes across graphs. Data are expressed as the mean ± standard error. Two-way ANOVA with post-hoc Bonferroni’s test.

**Figure 2 biomolecules-13-01107-f002:**
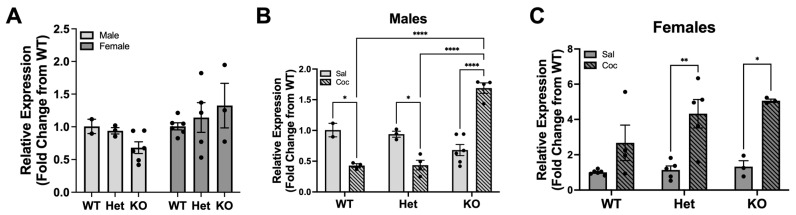
BRD4 gene expression levels in the PFC of DAT transgenic rats. (**A**) Basal level of BRD4 across sexes and genotypes (Saline Males: WT (*n* = 2), HET (*n* = 3), KO (*n* = 6); Saline Females: WT (*n* = 6), HET (*n* = 5), KO (*n* = 3); Cocaine Males: WT (*n* = 3), HET (*n* = 4), KO (*n* = 4); Cocaine Females: WT (*n* = 4), HET (*n* = 5), KO (*n* = 3)). Effect of cocaine on BRD4 gene expression across (**B**) male and (**C**) female rats. * *p* < 0.05, ** *p* < 0.01, **** *p* < 0.0001 represent significant differences. Note different y-axes across graphs. Data are expressed as the mean ± standard error. Two-way ANOVA with post-hoc Bonferroni’s test.

## Data Availability

Data are contained within the article and are available from the corresponding author on request.

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
