# Peer review of "Dopamine Transporter Knockout Rats Display Epigenetic Alterations in Response to Cocaine Exposure"

_biomolecules, 2023, doi:10.3390/biom13071107_

Round 1
Reviewer 1 Report
Thank you for submitting your work. Your article demonstrates sex disparities in epigenetic marker expression in the prefrontal cortex (PFC) following cocaine administration in rats. These findings are interesting and may have implications for the design of treatment options for cocaine-related disorders. However, it is worth noting that the sample size employed in the study was relatively small, which could potentially weaken the support for the claims made. Consequently, the generalizability of the findings may be limited. Additionally, the investigation of underlying mechanisms responsible for the observed sex disparities in epigenetic marker expression was absent, and the discussion lacked any guidance on future research directions. Nevertheless, there are several areas that could be addressed to enhance the persuasiveness of your report.
Kindly provide a comprehensive discussion on the dopamine (DA) levels in the presented cases or show data.
-
Could you clarify the rationale behind collecting tissues specifically after 60 minutes following the final dose administration?
-
It would be beneficial to explicitly state the number of animals used in each experiment within the figures.
-
Do the "saline" data in both panels B and C correspond to the data presented in panel A, in both Figures 1 and 2?
-
Could you please provide the p-value between WT and Het in males for Figure 1?
-
It would be advantageous to incorporate scatter plots overlaid on the bar graphs as a visual representation.
-
In both Figures 1 and 2, the fill pattern for the cocaine animal data in WT differs from that of Het and KO. Kindly make the necessary adjustment.
-
There appears to be a typographical error in line 153, where "increased" should be corrected.
Author Response
We thank so much your time reviewing our article and giving us great suggestions to improve our manuscript. We believe we have answered each of your inquiries but we would be more than welcomed to further discuss any additional suggestion you may have. Please see attached each answer to your comments.

Reviewer 2 Report
This manuscript identifies interesting sex differences in the relationship between DAT and epigenetic modifications following cocaine use. The present data expands on current data in males that indicate KDM6B and BRD4 expression are changed during extinction and withdrawal. In addition, the role of gene dosage effects using hetereozygous rats are examined. The authors conclude that inhibiting these epigenetic markers would benefit addiction outcomes. While these data are of interest, particularly for the sex differences identified, they require some further explanation and control studies to ensure that the conclusions are within the scope of the present dataset.
Abstract
In (3) results it is stated that “We show that only full knockout 19 (KO) of DAT impacts basal levels of KDM6B markers in a sex-dependent manner” It would be more clear to state that full KO only affects females.
The conclusion may be a stretch as no functional studies were performed. Both targets showed the same pattern of effect in sex differences, perhaps functionality in identifying sex specific targets might be more impactful.
Introduction
Line 60. References 28 and 29 do not involved addiction, this point should be separated to highlight that these references are either memory or addiction relevant.
Line 64. It is a stretch to say a potential target for treatment, as none of the referenced studies are functional manipulations. Perhaps that it is involved in CUD progression.
Line 80. The role of BDNF in the development of SUD (should be CUD, as SUD is not defined), is not clear. Does BDNF play a role in acquisition, extinction, reinstatement? Perhaps the connection to BDNF is weak, and instead could be left at its importance in cocaine exposure.
It is unclear why so much of the introduction focuses on memory, as none of the testing herein uses conditioned place preference or cocaine memory retrieval. While these factors are important for contextual retrieval events, the background contextual information should focus on results that are comparable to the present studies.
Methods
The total number of animals per sex and genotype and average weights should be reported. The background strain of rat should also be reported, and a citation for the prior studies using this rat strain in the Gainetdinov lab needs to be provided.
It is unclear why 8 days of non-contingent cocaine was used.
What is the justification for euthanasia at PND 42 instead of PND 55?
2 rats in a genotype group is insufficient. Were rats collected across litters or was this study performed in a single litter? Per the statistics reported below, ~36 rats were used total. This amounts to approximately 3 litters, indicating that if any group has less than 3 rats (per the statistics, some groups have 2), representative litter mates would not be possible. Is there a justification for why more litters were not included? Are these effects consistent within litters since the data is not equivalently representative?
What was the stereotaxic limits for PFC collection? What regions of the PFC dorsal, ventral, anterior, posterior, were included? What was the range in tissue weight? A figure that demonstrates the range of tissue collected would be helpful.
The average quality of RNA should be reported. Were any samples discarded for poor RNA extraction?
Results
All figures should be reported with individual data points and bars. In figure 1/2, it states n=2-6/group. 2 animals in any group is insufficient to draw conclusions. Is this group the one with the highest variability? The number of animals in each group should be readily available.
The authors state response to cocaine administration, however this is not entirely accurate. The authors tested the response to 8 days of cocaine, and then sacrificed rats 60 minutes following a cocaine injection. It is unclear whether these effects are due to repeated cocaine exposure or acute cocaine withdrawal. Either a group with a singular day of cocaine and the same 60m withdrawal period, or a group sacrificed with cocaine on board, would sufficiently answer this question.
Discussion
The authors assert that their results indicate that KDM6B is elevated by cocaine extinction. However, all comparisons are to cocaine naïve rats and all rats in this study were sacrificed during acute cocaine withdrawal. Extinction would need to be paired to a context. The terminology of extinction should be removed throughout and replaced with acute withdrawal.
A discussion of how expression of these factors in the PFC differ from expression in the NAc is necessary.
Overall it is difficult to interpret the present results in the context of prior findings, as it is unclear what results are statistically powered and which are not. Furthermore, the study is poorly controlled to derive the conclusions it is attempting to make. It is not clear whether these effects are withdrawal dependent, or dependent on cocaine exposures. In fact, prior reports in males indicate that these effects are consistent with acute withdrawal, and no necessarily from repeated cocaine. The manuscript needs to refine the specific effects they are attempting to identify instead of making broad strokes.
Conclusions.
While these studies indicate that these epigenetic enzymes are altered by cocaine, and dependent on DAT, there are no functional studies included. Since KDM6B and BRD4 are highly involved in memory consolidation, reconsolidation, and extinction, it is a stretch to suggest without any preclinical data that inhibiting them would promote abstinence. At most, they may be targets for future studies.
Figures.
# should not be used to indicate significant differences between groups, as it could be confused for trends. Use an alternative symbol.
Figure 1. It is unclear what the fold change comparisons are across graphs. In Fig 1A, are females being compared to WT males or WT females? Using 2 ranges separated by \\ on the Y axes in Fig 1C would help, so that we could see the same SAL effect as in A. Is the increased expression in male hets not significant due to an underpowered statistic? The error is very small. As we do not know which groups have 2 animals, it is impossible to decipher. Similar to the point in the results section, it is unclear why WT differences are not significant in C, as they are likely underpowered again. More animals are required to confirm that this is not a significant difference.
The effect in C may simply be a cocaine effect, independent of genetic condition. In the same reasoning, within each genetic condition, are fold change differences compared to saline the same fold change? These post-hoc tests should be performed to determine if the effects of cocaine are greater, or if it is simply a baseline difference across genetic conditions. Again, as the individual numbers of animals in conditions are not shown, it is impossible to tell what effects are possibly real versus underpowered.
Figure 2. Identical issues to above.
Other minor points:
The manuscript should be read for some writing errors.
Line 134 double space between enzyme and to
Statistics do not have consistent formatting.
Line 153, increase is misspelled.
SUD and CUD are used interchangeably, but SUD is never defined. As these rats are not self-administering drug, it might be more conservative to use SUD when discussing clinical relevance.
Author Response
We appreciate so much your time revising our manuscript. We thank your detailed suggestions. We believe we have answered each of your questions in the attached document. We would be pleased to further discuss any additional comments you may have after revising our resubmission. We have modified the document including, when pertinent, the answers to your comments and we believe the document has been strongly improved.
Thank you again for your input and time, is strongly appreciated.

Reviewer 3 Report
In the manuscript titled ‘Dopamine Transporter Knockout Rats display Epigenetic Alterations in 2 Response to Cocaine Exposure’, the authors found that chronic cocaine exposure caused strong epigenetic changes in rats in a sex and genotype-dependent manner. The results are very interesting. I only have few comments here.
1, Dopamine neurons project to different brain regions, like nucleus accumbens, amygdala, hippocampus, olfactory tubercle, and PFC. Since PFC only receives a small dopaminergic population input, but nucleus accumbens and amygdala receive a relative larger dopaminergic population input. Exam the epigenetic changes of NAc and amygdala would give this manuscript more solid support.
2, DAT KO may cause hyperactivity in animals. Did the authors notice such hyperactivity in KO rats? 10 mg/kg cocaine may increase the locomotor activity in WT rats, what would be the activity in KO rats after 10 mg/kg cocaine injection?
3, The expression of KDM6B was increased in females DAT KO rats, would the KDM6B inhibitor bring the cocaine-induced locomotor or KO-induced hyperlocomotion in females to the level of WT?
Author Response
We thank so much your comments and suggestions. We believe we have answered each of your inquiries. We will strongly take into consideration your suggestions to continue our project in future studies.
We thank again your input and time.
Sincerely,
Marta

Round 2
Reviewer 2 Report
I have no additional comments, the authors did a considerable job responding to comments and addressing concerns.
Fine
Reviewer 3 Report
The authors addressed all my concerns.